# The kinetics of carbon pair formation in silicon prohibits reaching thermal equilibrium

Peter Deák[1], Péter Udvarhelyi [1,2], Gergő Thiering[1] & Adam Gali [1,2] ✉

Thermal equilibrium is reached when the system assumes its lowest energy. This can be hindered by kinetic reasons; however, it is a general assumption that the ground state can be eventually reached. Here, we show that this is not always necessarily the case. Carbon pairs in silicon have at least three different configurations, one of them (B-configuration) is the G photoluminescence centre. Experiments revealed a bistable nature with the A-configuration. Electronic structure calculations predicted that the C-configuration is the real ground state; however, no experimental evidence was found for its existence. Our calculations show that the formation of the A- and B-configurations is strongly favoured over the most stable C-configuration which cannot be realized in a detectable amount before the pair dissociates. Our results demonstrate that automatized search for complex defects consisting of only the thermodynamically most stable configurations may overlook key candidates for quantum technology applications.

Computational materials science has the task to simulate the consequences of structural changes for the functional properties of materials. This is usually done by searching for the configuration with the lowest free energy (in solids most often for the lowest energy), and then calculating the observable properties. Therefore, such theoretical results pertain to thermal equilibrium. In real-life experiments, that cannot always be realized due to kinetic factors, which can, of course, also be modelled. However, it is expected, that under appropriate conditions all system will eventually reach thermal equilibrium, i.e., the state with lowest (free) energy. This is also assumed for complexes of point defects in semiconductors that may appear in various configurations. The most common approach for identification of observable defect complexes is to search for the energetically most favourable configuration either using molecular dynamics[1–3] or automatized algorithms of quasi-static total energy calculations[4,5]. The development of such algorithms has become very intense because fluorescent and paramagnetic point defects in solids may realize single photon sources and qubits, and there is an urgent quest to seek novel candidates optimized for the target quantum technology application[6]. In this paper we demonstrate for a single photon emitter in silicon[7–10] with an optically-read electron spin[11], that the basic assumption of thermodynamically stable configuration of defect complexes used

behind the search for novel quantum emitters is not granted generally. Our results could turn the direction of developments in the field towards considering the kinetic factors too, with using realistic parameters for the condition of formation, e.g., annealing temperature.

We exemplify this, in particular, on the so-called G photoluminescence (PL) centre[12] in silicon which emits in the telecom O band. A single photon source with telecom wavelength may turn silicon into a quantum-coherent material which unifies the electronics, photonics, and quantum optics components into a single, completely integrable platform. Accordingly, the G centre is receiving increased attention recently. In their seminal work, the group of G. D. Watkins[13] has identified the origin of the G centre as the neutral state of a dicarbon defect. In the ground state, termed B-configuration (Fig. 1B), the two carbon atoms are substitutional, with a buckled bond-centre silicon interstitial between them. The defect is bistable and transforms upon both positive and negative charging into the A-configuration (Fig. 1A), which is a (C-Si)$_{Si}$ pair on a lattice site (split interstitial), next to a substitutional carbon. With the application of several experimental techniques, the authors of this ingenious paper have worked out all charge transition levels of, and transformation barriers between these configurations with high accuracy. In the neutral charge state the B-configuration has an energy lower than the A-configuration by

---

[1]Wigner Research Centre for Physics, P.O. Box 49, H-1525 Budapest, Hungary. [2]Department of Atomic Physics, Institute of Physics, Budapest University of Technology and Economics, Műegyetem rakpart 3., 1111 Budapest, Hungary. ✉e-mail: gali.adam@wigner.hu

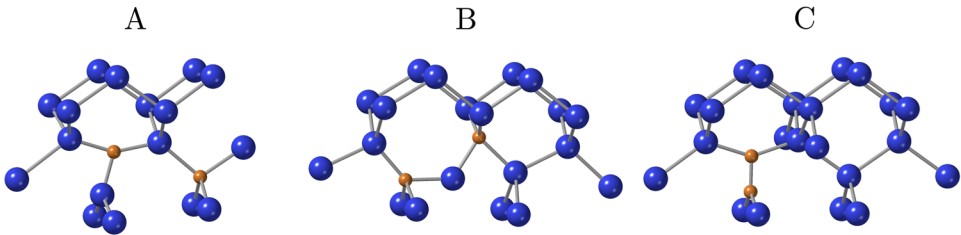

**Fig. 1 | Three configurations of carbon pair defects in silicon. A**, **B**, and **C** label the three configurations identified in prior studies. Brown and blues spheres are the carbon and silicon atoms, respectively. The figures are made by Crystal Maker™.

0.02 eV. The activation energy of transforming B into A-configuration is between 0.15 eV and 0.21 eV depending on the charge state. The (0/−) charge transition level in the A-configuration is 0.17 eV below the conduction band minimum (CBM), while the (+/0) level is 0.09 eV above the valence band maximum (VBM).

In the carbon pair in silicon has been investigated by electronic structure calculations as well[14–20]. It was found that the most stable configuration is a $(C–C)_{Si}$ split-interstitial pair, oriented along the <001> axis. However, this−so-called−C-configuration (Fig. 1C) has never been observed, while the G centre is associated with the less stable B-configuration[20]. Besides the G centre at 969 meV, the PL spectrum of carbon-containing irradiated silicon shows prominent lines at 856 meV, due to the $(C–Si)_{Si}$ split-interstitials, the C-centre at 789 meV, and the W-centre at 1018 meV[21,22]. The C-centre has been identified with a C-O defect (forming a ring structure)[23], and the W-centre with a self-interstitial complex $(I_3)$[10], recently. It was shown by GW calculations[19] that the electronic structure of the C-configuration is significantly different from that of the B-configuration (G centre), so it does not show up in the PL spectrum. In addition, the ground state of the C-configuration is triplet, while the optically detected magnetic resonance centre associated with the dicarbon defect exhibits a singlet ground state[13].

In this paper we investigate the formation of the various configurations and their possible transformation into each other by means of highly accurate theoretical methods. We show that in the temperature region, where the G centre exists, kinetic factors prevent the formation of the most stable C-configuration which reconciles the apparent contradiction with predicting the most stable C-configuration from theory against the observed B-configuration in experiments.

## Results and discussion

The three configurations in question for the dicarbon pair are shown in Fig.1, and the total energies, calculated with supercell plane wave density functional theory of HSE06 functional in the 512-atom supercell (see "Methods"), are listed in Table 1.

Since the HSE06 functional reproduces the band gap and satisfies the generalized Koopmans' theorem (gKT) in silicon to a good accuracy, it can yield very accurate results for defects (see Methods), although the close agreement of the calculated energy difference

between the A- and B-configuration (0.02 eV) with experiment[13] might be a bit fortuitous. Our calculations confirm that the spin-triplet state of C-configuration is the most stable configuration in the neutral charge state by far but even its spin-singlet state is favoured over B-configuration. C-configuration is also lower in energy than A-configuration in both the positive and in the negative charge states. Therefore, the question arises: why has C-configuration been never observed?

The G centre can appear after producing interstitial silicon atoms in carbon-containing samples (typically a few times $10^{16}$ cm$^{-3}$ in commercial Czochralski-silicon wafers and silicon-on-insulator samples). The moving silicon interstitials, in turn, produce mobile carbon $(C–Si)_{Si}$ split interstitials, the latter diffusing with an activation energy of 0.72–0.75 eV[24]. It is generally assumed that the dicarbon defect arises upon the encounter with a substitutional carbon (see e.g. ref. [19]). After high-dose carbon implantation, in principle, a dicarbon defect could also form with a different mechanism, namely at the encounter of two mobile carbon $(C–Si)_{Si}$ interstitials, leading to the emission of a silicon $(Si–Si)_{Si}$ split self-interstitial. We have calculated the energy of that reaction in the 512-atom cell with HSE06, and found it to be endothermic by 0.39 eV (see Supplementary Note 1). We anticipate that one of the reasons is that this reaction is unfavourable because the dicarbon defect contains a substitutional carbon, $C_{Si}$, and we find that the reaction of a carbon $(C–Si)_{Si}$ interstitial with the Si lattice forming a $C_{Si}$ and ejecting a silicon $(Si–Si)_{Si}$ split self-interstitial is strongly endothermic by 1.85 eV. Nevertheless, during the annealing most of the implanted carbon ions will go substitutional (when vacancies are available generated by the high-energy carbon ions) which is, according to our calculation, energetically more favourable than the interstitial by 2.29 eV. $C_{Si}$ is always neutral (see Supplemental Note 1 for the electronic structures of defects) and its tensile stress field will attract the carbon $(C–Si)_{Si}$ interstitials, which exert a compressive stress on the environment. Therefore, it is to be expected that the formation mechanism of the G centre in carbon implanted pure samples is the same as in the case of irradiated carbon containing samples.

So we have investigated the scenario of $(C–Si)_{Si}$ encountering $C_{Si}$. If the barrier for a reaction between them is higher for the formation of the C-configuration than for A-configuration or B-configuration, then the latter two (energetically very close) configurations will be dominantly formed, and if the barrier for transforming A-configuration or B-configuration into C-configuration is high, the absence of experimental evidence for C-configuration can be explained.

At the HSE06 level, the 512-atom unit is way too large for minimum energy path (MEP) calculations, so we used the 64-atom supercell for that (see "Methods"). Carbon $(C–Si)_{Si}$ interstitial diffuses by a reorientation mechanism[25]. We have recreated this diffusion path in the 64-atom supercell and obtained 0.70 eV for the activation energy. The good agreement with the experimental value shows that the computational parameters are appropriate. According to experiment, the A-configuration is the stable one in both the positive and the negative charge state, and even in the neutral one, the slightly more

**Table 1 | Relative energies, of the three carbon-pair configurations in the 512-atom supercell, for the neutral and the charged states, with respect to the neutral _B_-configuration**

| HSE06 | $E$ (eV) | | $E^+ + E_F$ (eV) | | $E^- - E_F$ (eV) |
|---|---|---|---|---|---|
| $^1$A | 0.02 | $^2$A(+) | 0.50 | $^2$A(−) | 0.33 |
| $^1$B | 0.00 | | | | |
| $^1$C | −0.16 | $^2$C(+) | 0.13 | $^2$C(−) | −0.03 |
| $^3$C | −0.48 | | | | |

The Fermi level, $E_F$, is assumed to be at midgap. The spin state $(2S+1)$ is indicated in the left superscript.

stable B-configuration can easily be transformed into A-configuration, with an activation energy of only 0.16 eV[6], i.e., much lower than the diffusion activation energy. Therefore, we have investigated and compared the formation of the A- and C-configurations when $(C-Si)_{Si}$ encounters $C_{Si}$, as well as the transformation of A-configuration into C-configuration. We have considered A-configuration, as the final and the initial configuration, respectively, both in the neutral and the positive charge state.

As can be seen in Fig. 2, the A-configuration arises during the motion of $(C-Si)_{Si}$ with the normal reorientation mechanism. The formation of the C-configuration is a more complex process, involving two simultaneous bond-switches as shown in Fig. 3.

The variation of the total energy (with respect to the initial state) is shown in Fig. 4. The energy barrier for the dissociation of the A-configuration into $(C-Si)_{Si} + C_{Si}$ (reverse direction on Fig. 4a) is 1.86 eV, in excellent agreement with the experimentally observed activation energy of 1.90 eV, with which the G centre anneals out[26]. A PBE estimate of the contribution of harmonic vibrations to the Helmholtz free energy barrier was obtained at 100 °C (see Methods and Supplemental Note 2). It has lowered the barrier energy only by 0.13 eV which is within 7% w.r.t the calculated HSE06 barrier energy at zero kelvin without zero-point energy corrections.

The activation energy of the last step in the formation of the dicarbon pair is -0.3 eV in both charge states for the A-configuration

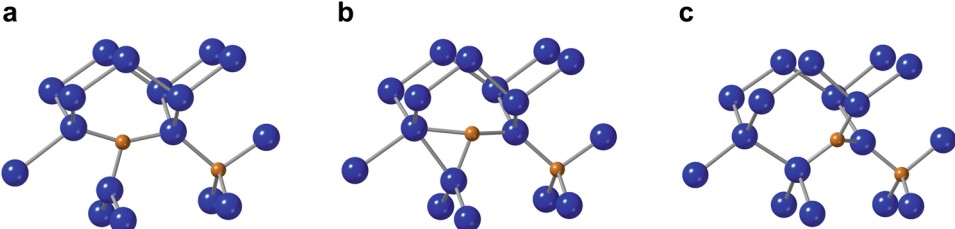

**Fig. 2 | Path for the formation of the A-configuration upon the encounter of a diffusing $(C-Si)_{Si}$ split interstitial and a $C_{Si}$ substitutional. a** initial state, **b** (near-)saddle-point configuration (c.f. Fig. 4), and **c** end state of the MEP calculation.

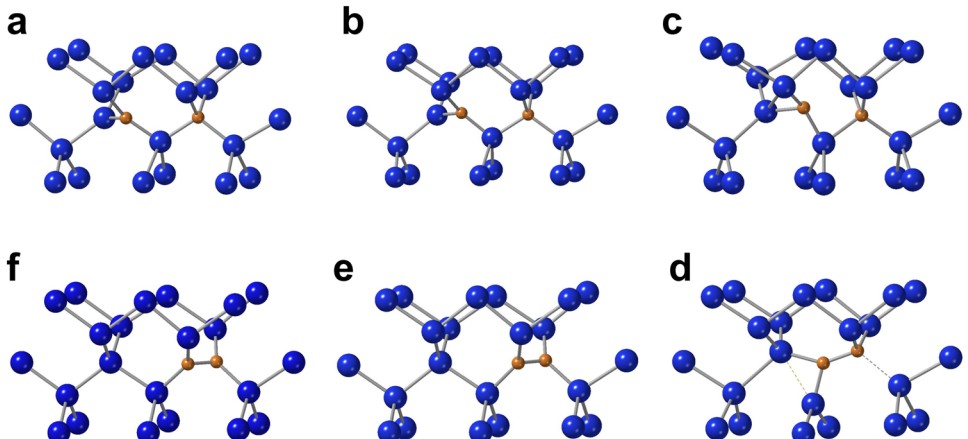

**Fig. 3 | Path for the formation of the C-configuration upon the encounter of a diffusing $(C-Si)_{Si}$ split interstitial and a $C_{Si}$ substitutional. a–f** The calculated configurations along the reaction coordinate. Dashed lines indicate bonds in the breaking/forming, and are just guides to the eye.

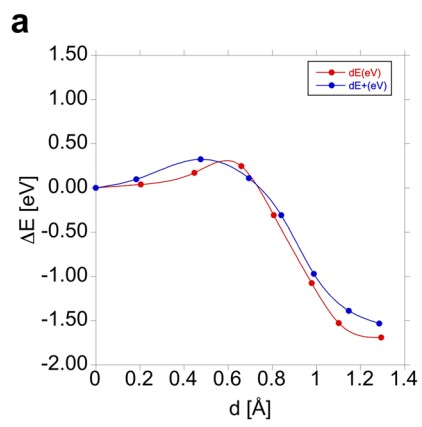
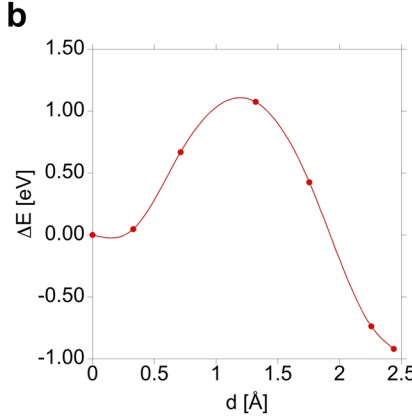

**Fig. 4 | The change of the total energy with respect to the initial state, as a function of the distance travelled by the C-interstitial when a $(C-Si)_{Si}$ split-interstitial approaches a $C_{Si}$ substitutional.** The curves are cubic spline fits, to guide the eye. **a** The formation of the neutral (red curve) and positive (blue curve) A-configuration. **b** The formation of the neutral C-configuration.

(note that the MEP does not change much with the number of electrons, and that can be expected even if adding one more) and it is 1.2 eV for the C-configuration. The relation of these barriers corresponds to a relation of 30–40 of the formation rates between room temperature and 100 °C, where the G centres are typically created, achieving a concentration of $<1 \times 10^{16}$ cm$^{-3}$ (see ref. [19] and references therein). This means that the expected concentration of carbon pairs in the C-configuration is $<3 \times 10^{14}$ cm$^{-3}$, i.e., below the detection limit of ensemble measurement techniques. So, from the practical point of view, the more stable C-configuration does not form.

In principle, the A- (and B-) configuration could still transform into C-configuration upon further annealing at higher temperatures, so we have considered this transformation. The variation of the energy with respect to that of the A-configuration is shown in Fig. 5. The transformation to C-configuration is a very complicated process, going through a local minimum before, and a stationary state after the saddle point (as shown in Fig. 6). The activation energy is way higher than the one required for the A-configuration to dissociate (1.9 eV), so this transformation will never happen. Considering the <1.2 eV gap of silicon, the A- and B-configurations will be maintained even upon non-radiative recombination of an electron-hole pair after optical excitation.

We have shown that the activation energy for the formation of the most stable C-configuration of the carbon pair defect in silicon (1.2 eV) is substantially higher than the one required to form the A-configuration (0.3 eV), so the latter (and the B-configuration into which A-configuration can easily transform) will dominantly occur, and the more stable C-configuration will be below the detection limit. The activation energy to transform A- (and B-) configuration into the C-configuration is also much higher than the one required to anneal

out the carbon-pair defect, and cannot even be surmounted with the help of non-radiative recombination of an electron-hole pair. This could explain the long-lasting stability of the G centre under intense above-band-gap illumination (e.g. ref. [27]).

In summary, we demonstrated that, in the case of complexes, the thermodynamically most stable configuration of the defect may never form in detectable amount, irrespective of the experimental conditions. In some cases, configurations of higher energy are preferred kinetically because the complex breaks up before being transformed from these into the lowest energy configuration. This underlines the importance of simulating the transformation of different possible defect configurations into each other, especially if studying processes occur in irradiated/implanted materials.

## Methods

Calculations were carried out with the Vienna ab-initio simulation package VASP 6.2, using the projector augmented wave method[28–30], and a plane wave cutoff of 420 (840) eV for the wave function (charge density). The Heyd-Scuseria-Ernzerhof (HSE) hybrid functional[31] was used with mixing parameter $\alpha = 0.25$ and screening parameter $\mu = 0.20$, since this (so-called HSE06) parametrization[32] produces a band gap of 1.16 eV (in excellent agreement with the low temperature single-particle gap of 1.17 eV)[33], and satisfies the generalized Koopmans' theorem (gKT) in silicon[34], which are the two conditions for accurate results on defects[35]. Defects were modelled in a supercell, and all atoms were allowed to relax in a constant volume till the forces were below 0.02 eV/Å. Activation energies were determined by using the nudged elastic band (NEB) and climbing image NEB (CI-NEB) methods[36–38].

The relative stabilities of the three different configurations of the carbon pair (A, B, and C) have been established in a 512-atom supercell ($4 \times 4 \times 4$ multiple of the Bravais-cell), while migration paths were studied and the activation energies calculated in a 64-atom ($2 \times 2 \times 2$) cell, always using the Γ-point approximation for Brillouin-zone sampling. The lattice constant was taken from our earlier HSE06 work[33] to be 5.4307 Å (in good agreement with experiment).

Total energies of charged 512-atom supercells have been calculated by applying the self-consistent potential correction (SCPC)[39]. In studying the minimum energy path (MEP) of the formation and transformation of the various dicarbon configurations, we are interested in relative energies only, and since the localization of the charge does not change significantly between the configurations, no charge correction was applied.

The contribution of harmonic vibrations to the Helmholtz free energy barrier was calculated in the 64-atom cell, using the Perdew–Burke–Ernzerhof (PBE)[40] functional and allowing all atoms to vibrate.

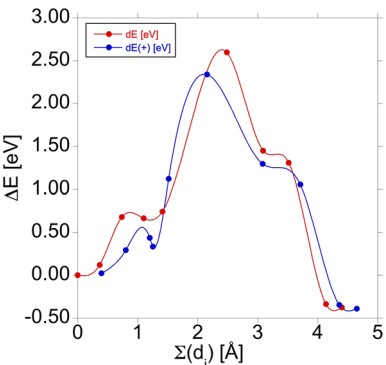

**Fig. 5 | The change of the total energy during the transformation of the A-configuration into the C-configuration in the neutral (red curve) and in the positive (blue curve) charge state.** The configuration coordinate is the sum of the distances the interstitial carbon atom travels in the steps of the transformation.

## Data availability

The authors declare that the main data supporting the findings of this study are available within the paper. The data that support the findings

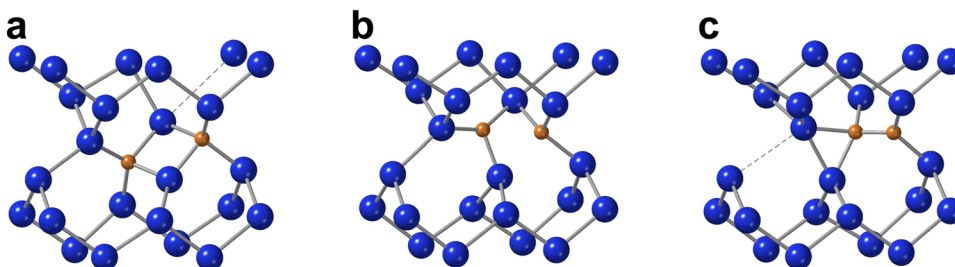

**Fig. 6 | The path of the transformation of the A-configuration into the C-configuration (c.f. Fig. 5).** Dashed lines indicate bonds in the breaking/forming, and are just guides to the eye. **a** The local minimum. **b** The saddle point. **c** The stationary state.

of this study are available from the corresponding author upon reasonable request. Source data are provided with this paper.

## Code availability
The authors used the standard VASP 6.2 package for density functional calculations that is an ab-initio software package with license fee.

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

## Acknowledgements

Support by the National Excellence Program for the project of Quantum-coherent materials (NKFIH Grant No. KKP129866) as well as by the Ministry of Culture and Innovation and the National Research, Development and Innovation Office within the Quantum Information National Laboratory of Hungary (Grant No. 2022-2.1.1-NL-2022-00004) is much

appreciated. We acknowledge the high-performance computational resources provided by KIFÜ (Governmental Agency for IT Development) institute of Hungary. Open access funding provided by ELKH Wigner Research Centre for Physics.

## Author contributions

P.D. and P.U. carried out the DFT calculations under the supervision of A.G. P.D. and G.T. analyzed the results under the supervision of A.G. All authors contributed to the discussion and writing the manuscript. A.G. conceived and led the entire scientific project.

## Funding

## Competing interests

The authors declare no competing interests.
