## [Peer Review File · Nature Communications]

REVIEWER COMMENTS

Reviewer #1 (Remarks to the Author):

The authors have performed several density functional calculations of the formation processes of carbon pair formation in silicon, using an hybrid exchange correlation functional and the Nudged Elastic Band method. The validity of these calculations are first assessed by comparison with the literature. The authors then investigated the formation of the three possible carbon pair configurations A, B and C, starting from a substitutional carbon and a split C-Si interstitial, and concluded that the most stable C required a higher energy than metastable A or B. In addition, they also found that formation energy of C from B or A is higher than their dissociation. They then conclude that this most stable C state can never be obtained.

These calculations appear to be carefully done, and the manuscript is short and pleasant to read, and convey efficiently the main message. These results are probably interesting enough to warrant publication in Nature Communication. However, I have a few questions and requests that I feel must be correctly addressed before publication, in my opinion:

1) I am not knowledgeable enough of carbon pairs in silicon so this is maybe a naive question. The authors explained that configuration C has never been observed. But is this configuration easily observable? Does it have a clear optic or electronic signature ? The authors should clarify this point in the text.

2) There is always the question of finite temperature contributions, which are not considered in this work. The reported energy differences are large, then I do not expect huge changes for temperatures close to 100 °C, but I think that the authors should comment in their manuscript about potential harmonic and anharmonic contributions to free energy. Furthermore, even if C atoms induce low distortions in silicon, the use of a small 64 atoms cell should be discussed too.

3) The last issue is more fundamental and major in my opinion. The authors are very unequivocal in their conclusion, with the use of expressions like 'never form' or 'prohibited'. I would like to be as convinced as they are. Did the authors have considered/investigated all the possible paths for the formation of the C pair? They showed a mechanism where the initial reactants are a substitutional C and a split C-Si interstitial. Why such a unique choice? What if two split C-Si interstitials are used instead? How can the authors be sure that they explored all possible mechanisms for C formation? Besides did they consider that a C pair could form from the encounter of A (or B) configuration with a third C atom (such as $A + C-Si \rightarrow C + C-Si$)? Finally, is it possible that the charge locally change during the migration mechanism (It seems to me that this possibility has been put forward by some authors, but I do not remember the publication), thus eventually helping the C formation? I believe the authors should add a paragraph in their manuscript to discuss all these aspects.

Reviewer #2 (Remarks to the Author):

In this research the authors theoretically demonstrate the difficult kinetic access to the lowest energy configuration for the carbon interstitial carbon substitutional CI-CS complex (pair) in silicon. The metastable CI-C configurations in Si are of interest for quantum technologies and their higher formation yield, inferred by this computational analysis, justify the experimental evidences. The computational schemes applied are highly accurate for the evaluation of the configurations' energetics and the related electronic-spintronic structure. The great difference of the complex formation barrier starting from the separate CI and CS is correlated to consistence of the formation mechanism for the metastable configurations $A \Rightarrow B$ with the diffusion mechanism on the CI mobile specie, whilst the stable complex requires the positions' inversion of the Si C atoms in the diffusing CI. This inversion should be the high energy event for the eventual formation split C-CI (C configuration). In my opinion computational methodologies applied are extremely accurate and the rationale of the research is clear and well presented, moreover the overall results are sound and interesting for a large audience since the main concepts can be qualitatively extended to the evolution of a broad class of atomic structures. My appraisal is that this research could be

published in Nature Communications, although some points hereby outlined need further refinements/discussions.

I believe that in addition to other values, characterizing the energetics of the complex, also the barrier related to the inversion of the C and Si atoms in the isolated CI, i.e. the (C-Si)₂Si specie, should be calculated. This result could be useful for the overall discussion.

Several configurations are shown with bonds indicated in the stick and ball representation. I would imagine that bond assignment follows some rigorous rule related to the local electron density.

Could you clarify this issue? What is the meaning of bonds indicated by dashed segments?

In principle also the encounter of two CI-CI mobile species could lead to the CI-CS complexes and the release of a mobile Si self-I. There it the possibility that a large part of CS converts in CIs in presence of a large I super- saturation. One wonders if the same or similar structural selection rule which hinder the lowest split like configuration applies for this hypothetical CI-CI formation path. Could you discuss this issue?

Probably raw data reporting the atom positions in the computational cell, at least for key configurations, could be reported as supplementary material.

Reviewer #3 (Remarks to the Author):

The workflow provided by Authors to clarify the experimental observation of G photoluminescence center in monocrystalline silicon and to demonstrate why the lowest energy structure of the defect (C-C <001> split interstitial) cannot be created in detectable concentration is correct. First, the total energies of different possible configurations of C-C defect were determined using hybrid HSE06 exchange-correlation functional and large 512 atomic supercell. Next, the Authors used CI-NEB method developed by Henkelman group to map the potential energy surface for C-C defect transformations and determine the minimum energy pathway (MEP) and the activation barriers. That approach was sufficient to reproduce the available experimental data with good precision. Although from technical point of view the paper does not have any major flaws the novelty of the research is way too low for Nature Communications. The reasoning behind my statement is following:

- 1) The impact on the defect-related physics, and in particular, quantum information science is negligible. In fact, the Authors presented only one example of such behavior and it can be dangerous to draw any general conclusions as we do not know if this behavior is ubiquitous in other systems.
- 2) There are numerous examples of physical systems (not only point defects) that exhibit a complex potential energy surface (PES) with many local minima, saddle points and valleys. It is very obvious that if at least one of these local minima is deep enough (activation barrier is relatively high) the system can get stuck in local minimum without reaching global minimum.
- 3) There is a lack of novelty when it comes to methodology. The Authors used very well-established and standard methods to perform the research. It is nothing wrong with using standard computational tools if the research itself carries a significant impact. However, for Nature Communication publication the methodology must be novel, in particular, when the impact of the research itself is negligible or moderate.
- 4) I am surprised that the Authors did not even discuss the configurational and vibrational entropy contributions to the Gibbs free energy. In fact, these contributions can be significant (the order of 0.2 – 0.5 eV) between various defects (for example Cu in silicon) and at high enough temperature may change the thermodynamic picture. It is also possible that the entropy contributions are very similar for the different structures of C-C defect and cancel each other out when the relative stability is considered. This issue need to be addressed.

RESPONSE TO THE REVIEWERS' COMMENTS

Reviewer #1

We thank the Reviewer for the positive assessment of our paper and the suggestions for improving it. Below we provide our response in blue ink and list the changes made in the MS in red ink.

1) I am not knowledgeable enough of carbon pairs in silicon so this is maybe a naive question. The authors explained that configuration C has never been observed. But is this configuration easily observable? Does it have a clear optic or electronic signature ? The authors should clarify this point in the text.

In response to the Referee's comment we added the following, after sentence stating that the C-configuration has not been observed. As the C-configuration ground state is a triplet, it can be principally observed by electron spin resonance techniques but rather the B-configuration was observed in experiments. Previous GW+BSE calculations on C-configuration imply (Ref. 19 in the revised MS) that it might have optical fingerprints but the calculated vertical excitation energy is at much longer wavelength (so the zero-phonon-line should be even at longer wavelength) than that reported for carbon related defects. We added the paragraph below to the main text:

Besides the G-centre at 969 meV, the PL spectrum of carbon-containing irradiated silicon shows prominent lines at 856 meV, due to the (C-Si)_{si} split-interstitials, the C-centre at 789 meV, and the W-centre at 1018 meV. The C-centre has been identified with a C-O defect (forming a ring structure), and the W-centre with a self-interstitial complex (I₃), recently. It was shown by GW calculations that the electronic structure of the C-configuration is significantly different from that of the B-configuration (G-centre), so it does not show up in the PL spectrum. In addition, the ground state of the C-configuration is triplet, while the optically detected magnetic resonance centre associated with the dicarbon defect exhibits a singlet ground state.

2) There is always the question of finite temperature contributions, which are not considered in this work. The reported energy differences are large, then I do not expect huge changes for temperatures close to 100 °C, but I think that the authors should comment in their manuscript about potential harmonic and anharmonic contributions to free energy. Furthermore, even if C atoms induce low distortions in silicon, the use of a small 64 atoms cell should be discussed too.

Concern about the effect of finite temperature was also raised by Referee 3, so we have made a PBE estimate of the contribution of harmonic vibrations to the Helmholtz free energy at 100°C (where the signal of interstitial carbon disappears and the G-center forms), at the geometry of the A-configuration and at that of its dissociation barrier. The results show that the barrier energy with vibration entropy contributions is lowered by 0.13 eV. This confirms the assumption of the Referee

1 above. Since anharmonic effects can hardly play a role at such low temperatures, using the 0K energies seems to be a good approximation. In the revised MS, we explicitly mention this when comparing the dissociation barrier of the A-configuration to the experimentally observed value.

A PBE estimate of the contribution of harmonic vibrations to the Helmholtz free energy barrier was obtained at 100°C (see Methods and Supplemental Note 2). It has lowered the barrier energy only by 0.13 eV which is within 7% w.r.t the calculated HSE06 barrier energy at zero kelvin without zero-point energy corrections.

As for the quality of the results from a 64-atom calculation, that was actually discussed already in the original MS:

1) Bottom of Page 4 in the original MS: “We have recreated this diffusion path in the 64-atom supercell and obtained 0.70 eV for the activation energy. The good agreement with the experimental value shows that the computational parameters are appropriate.”

2) Top of Page 6 in the original MS: “The energy barrier for the dissociation of the A-configuration into $(\text{C-Si})_{\text{Si}} + \text{C}_{\text{Si}}$ (reverse direction on Fig.4a) is 1.86 eV, in excellent agreement with the experimentally observed activation energy of 1.90 eV, with which the G-center anneals out.”

We believed that these two results justified the use of the 64-atom supercell, especially since – as the Referee pointed out too – the critical energy differences are between 0.5 – 1.0 eV, which could hardly be affected significantly by the relaxation beyond the 4th neighbor shell.

Nevertheless, we provide additional data in the Supplemental Note 1 where we compare the binding energy of the reactions associated with the formation of dicarbon defects as obtained in 512-atom and 64-atom supercells. The results agree within 0.1 eV. We think this result supports to apply 64-atom supercell calculations for barrier energies of reactions.

3) The last issue is more fundamental and major in my opinion. The authors are very unequivocal in their conclusion, with the use of expressions like 'never form' or 'prohibited'. I would like to be as convinced as they are. Did the authors have considered/investigated all the possible paths for the formation of the C pair? They showed a mechanism where the initial reactants are a substitutional C and a split C-Si interstitial. Why such a unique choice? What if two split C-Si interstitials are used instead? How can the authors be sure that they explored all possible mechanisms for C formation? Besides did they consider that a C pair could form from the encounter of A (or B) configuration with a third C atom (such as $\text{A} + \text{C-Si} \rightarrow \text{C} + \text{C-Si}$)?

Let's start with the reaction $(\text{C-Si})_{\text{Si}} + (\text{C-Si})_{\text{Si}} \rightarrow (\text{C-C})_{\text{Si}} + (\text{Si-Si})_{\text{Si}}$, since Referee 2 has asked the same question. Originally we omitted this possibility, since *i*) $(\text{C-Si})_{\text{Si}}$ may be positively charged, in particular, in *p*-type samples alike, i.e., would repel each other, whereas the dicarbon defect has been observed in such samples (therefore, we considered the positively charged carbon interstitial

too in our paper), and *ii*) the compressive stress, exerted on the environment would make it hard for them to approach each other when they are neutral.

In response to the comment of the Referees, we have now also calculated the energy of the

reaction, and have found it to be endothermic by 0.39 eV. So this possibility can definitely be ruled out. We note that even if the G-centers are created by carbon-implantation, during annealing most of the carbon will go substitutional (which is, according to our calculation 2.29 eV more favorable than the interstitial). The substitutional is always neutral and exerts tensile stress on the environment. Therefore, the C_i meets C_{Si} scenario is most likely even in this case. If the G-center has been produced in carbon containing samples with say silicon implantation or electron irradiation, the C_i meets C_i scenario would be out of the question anyway.

However, we thank the Referee for pointing out that these considerations have to be part of the paper for the general readership. Therefore, we have extended the second paragraph after Table I as follows:

The G-centre can appear after producing interstitial silicon atoms in carbon containing samples (typically a few times 10^{16} cm^{-3} in commercial Czochralski-silicon wafers and silicon-on-insulator samples). The moving silicon interstitials, in turn, produce mobile carbon $(C-Si)_{Si}$ split interstitials, the latter diffusing with an activation energy of 0.72-0.75 eV. It is generally assumed that the dicarbon defect arises upon the encounter with a substitutional carbon (see e.g. ref. **[Error! Bookmark not defined.]**). After high-dose carbon implantation, in principle, a dicarbon defect could also form with a different mechanism, namely at the encounter of two mobile carbon $(C-Si)_{Si}$ interstitials, leading to the emission of a silicon $(Si-Si)_{Si}$ split self-interstitial. We have calculated the energy of that reaction in the 512-atom cell with HSE06, and found it to be endothermic by 0.39 eV (see Supplementary Note 1). We anticipate that one of the reasons is that this reaction is unfavourable because the dicarbon defect contains a substitutional carbon, C_{Si} , and we find that the reaction of a carbon $(C-Si)_{Si}$ interstitial with the Si lattice forming a C_{Si} and ejecting a silicon $(Si-Si)_{Si}$ split self-interstitial is strongly endothermic by 1.85 eV. Nevertheless, during the annealing most of the implanted carbon ions will go substitutional (when vacancies are available generated by the high-energy carbon ions) which is, according to our calculation, energetically more favourable than the interstitial by 2.29 eV. C_{Si} is always neutral (see Supplemental Note 1 for the electronic structures of defects) and its tensile stress field will attract the carbon $(C-Si)_{Si}$ interstitials, which exert a compressive stress on the environment. Therefore, it is to be expected that the formation mechanism of the G-centre is the same as in the case of irradiated carbon contaminated samples.

So we have investigated the scenario of $(C-Si)_{Si}$ encountering C_{Si} . If the barrier for a reaction between them is higher for the formation of the C-configuration than for A or B, then the latter two

(energetically very close) configurations will be dominantly formed, and if the barrier for transforming *A* or *B* into *C* is high, the absence of experimental evidence for *C* can be explained.

Of course, the Referee is also right in saying that the number of theoretical possibilities are countless, and one cannot possibly investigate all of them. Summary formulas for reactions like “*A* + C-Si → *C* + C-Si” may be considered but looking at the actual structure shows its problems. Although it is highly unlikely (see the electric and stress field arguments above), let’s assume that another interstitial carbon, i.e. (C-Si)_{si}, could get close to *A* (which is the complex [(C-Si)_{si} + C_{si}]) in the most favorable way, as the lhs figure below shows. Then the transformation to the *C*-configuration would require the breaking of a C-Si bond (~ 4 eV) first (see rhs figure), so that is sure not going to happen. If the (C-Si) on the right of the lhs figure came in with Si down and C up, the transformation would require many simultaneous bond switching, yielding a barrier certainly not less than that of the *A* → *C* transformation.

In a similar manner, one can intuitively exclude many possibilities. Still, we (or anybody) could certainly not say to have investigated all of them. There are, however, clues from experiment, which all point to the mechanism of “C_i meets C_{si}”, as outlined above, and that’s why that is assumed generally in the literature.

Finally, is it possible that the charge locally change during the migration mechanism (It seems to me that this possibility has been put forward by some authors, but I do not remember the publication), thus eventually helping the *C* formation?

The change of the charge state during migration, i.e., the Burgoin-Corbett diffusion mechanism, occurs when the potential energy surfaces (PES) of two charge states cross each other, and the switch allows a lower barrier. The relation of the two PES depends on the position of the Fermi-level (E_F). In Figs. 4.a and 5 we have assumed *p*-type doping, with $E_F = \text{VBM} + 0.33 \text{ eV}$ and $E_F = \text{VBM} + 0.17 \text{ eV}$, respectively, in order to align the PES of the neutral and positive charges states at the initial configurations. As can be seen, at these particular Fermi levels, the PES of the two charge states run almost parallel, and a switch between them would not produce any significant gain in the barrier energy. In case of heavier *p*-type doping, the PES of the (+) charge state will become clearly preferred along the whole route, while for *n*-type doping that of the neutral. Therefore, our results seem to exclude Burgoin-Corbett diffusion. We think, however, that the

discussion above would not be of interest to the general readership of the journal, while experts in the field could deduce it by looking at Figs. 4a and 5.

Reviewer #2

We thank the Referee for the praise of our work and the suggestion for its publication. Below we provide our response to the particular comments and list the changes made in the MS.

I believe that in addition to other values, characterizing the energetics of the complex, also the barrier related to the inversion of the C and Si atoms in the isolated CI, i.e. the (C-Si)_{Si} specie, should be calculated. This result could be useful for the overall discussion.

We give an additional information in the revised MS that the C_{Si} defect is significantly more stable than the carbon (C-Si)_{Si} interstitial, by about 2.29 eV at HSE06 level of accuracy (see also below and the reply to Reviewer 1 above). As a consequence, the (Si-Si)_{Si} + C_{Si} → (C-Si)_{Si} reaction is indeed exothermic by 1.85 eV, thus mobile Si interstitials created by irradiation/implantation may kick-out C_{Si}, and (C-Si)_{Si} can form. On the other hand, this process occurs in the event of implantation/irradiation when the interstitial ions bear kinetic energy, thus the process is far from quasi thermal equilibrium, in contrast to annealing. In non-irradiated clean samples, the high-energy (Si-Si)_{Si} do not exist because they diffuse out from Si samples even at cryogenic temperatures. In this context, we think that knowing the barrier energy of the reaction from highly costly *ab initio* calculations would not alter the conclusions of our study. On the other hand, this reaction shows that carbon interstitials do not form substitutional carbon defects with ejecting Si interstitials, which has an indirect connection to the question below whether two approaching carbon interstitials could form a dicarbon defect with ejecting a Si interstitial. As expected from the highly endothermic nature of the reverse reaction for single carbon interstitial above, the answer is no. We think that the Reviewer's note really helped us to improve the MS with discussing this issue in detail (see below).

Several configurations are shown with bonds indicated in the stick and ball representation. I would imagine that bond assignment follows some rigorous rule related to the local electron density. Could you clarify this issue? What is the meaning of bonds indicated by dashed segments?

As indicated in the caption of Fig.1, the structures were drawn with the program CrystalMaker, which – as most similar tools – draws bonds based on standard radii: in our case the covalent radii of the atoms. The dashed segments indicate that a bond is about to break or about to be formed (based on the sum of the radii), so they just serve as a guide to the eye. Of course, we should have stated this, and did so now in the revised version, by adding the following to the figure caption.

Dashed lines indicate bonds in the breaking/forming, and are just guides to the eye.

In principle also the encounter of two CI-CI mobile species could lead to the CI-CS complexes and the release of a mobile Si self-I. There is the possibility that a large part of CS converts in CIs in presence of a large I super- saturation. One wonders if the same or similar structural selection

rule which hinder the lowest split like configuration applies for this hypothetical CI-CI formation path. Could you discuss this issue?

As described also in our answer to Referee 1, who asked the same question, we should point out first that in *p*-type samples the carbon interstitials are charged, so Coulomb-repulsion prevents them to meet and form complexes. Their encounter is hindered even in insulating samples (where they are neutral), since both induce a strong compressive deformation on their environment. As our calculation shows, the formation of (C-C)_{Si} from two (C-Si)_{Si} – with emission of (Si-Si)_{Si} – is actually exothermic by 0.39 eV. We also think that during annealing C_i will convert into C_s, and not vice versa, since the latter is favored energetically by 2.29 eV. We must agree though that such a discussion would have been important for our paper, and we thank the Referee for pointing that out. We have modified the second paragraph after Table I as follows.

The *G*-centre can appear after producing interstitial silicon atoms in carbon containing samples (typically a few times 10¹⁶ cm⁻³ in commercial Czochralski-silicon wafers and silicon-on-insulator samples). The moving silicon interstitials, in turn, produce mobile carbon (C-Si)_{Si} split interstitials, the latter diffusing with an activation energy of 0.72-0.75 eV. It is generally assumed that the dicarbon defect arises upon the encounter with a substitutional carbon (see e.g. ref. **[Error! Bookmark not defined.]**). After high-dose carbon implantation, in principle, a dicarbon defect could also form with a different mechanism, namely at the encounter of two mobile carbon (C-Si)_{Si} interstitials, leading to the emission of a silicon (Si-Si)_{Si} split self-interstitial. We have calculated the energy of that reaction in the 512-atom cell with HSE06, and found it to be endothermic by 0.39 eV (see Supplementary Note 1). We anticipate that one of the reasons is that this reaction is unfavourable because the dicarbon defect contains a substitutional carbon, C_{Si}, and we find that the reaction of a carbon (C-Si)_{Si} interstitial with the Si lattice forming a C_{Si} and ejecting a silicon (Si-Si)_{Si} split self-interstitial is strongly endothermic by 1.85 eV. Nevertheless, during the annealing most of the implanted carbon ions will go substitutional (when vacancies are available generated by the high-energy carbon ions) which is, according to our calculation, energetically more favourable than the interstitial by 2.29 eV. C_{Si} is always neutral (see Supplemental Note 1 for the electronic structures of defects) and its tensile stress field will attract the carbon (C-Si)_{Si} interstitials, which exert a compressive stress on the environment. Therefore, it is to be expected that the formation mechanism of the *G*-centre is the same as in the case of irradiated carbon contaminated samples.

Probably raw data reporting the atom positions in the computational cell, at least for key configurations, could be reported as supplementary material.

Thank you for the suggestion, we have done so as Supplemental Data.

Reviewer #3

Although from technical point of view the paper does not have any major flaws the novelty of the research is way too low for Nature Communications.

We thank the Referee for devoting time to assess our paper, however – with due respect – we must strongly argue the final judgement. We outline our views below, point-by-point, following the Referee's comments.

We have to start with comment No.2, which – we believe – is the main reason for the rejection, but is based on a misunderstanding.

2) There are numerous examples of physical systems (not only point defects) that exhibit a complex potential energy surface (PES) with many local minima, saddle points and valleys. It is very obvious that if at least one of these local minima is deep enough (activation barrier is relatively high) the system can get stuck in local minimum without reaching global minimum.

The problem we are facing here is not just the bumpy PES of an otherwise stable species but that of several complexes, which may diffuse, transform into each other, or dissociate. As also stated in the first paragraph of the Introduction, our conclusion is not equivalent to saying that “the given complex cannot get out of a local minimum below a certain temperature” but to “when the temperature is high enough to get out of the local minimum, the complex ceases to exist”, which is something quite different. We are not aware, that such a scenario had been proposed or shown before, and the existing textbooks about this topic should be adjusted, according to the results of this study. In response to this comment, however, we have reformulated the conclusion of our paper, to make our claim clear, as follows.

In summary, we demonstrated that, in the case of complexes, the thermodynamically most stable configuration of the defect may never form in detectable amount, irrespective of the experimental conditions, since lower energy configurations are preferred kinetically, and the complex breaks up before being transformed into the lowest energy configuration. This underlines the importance of simulating the transformation of different possible defect configurations into each other, especially if studying processes occurring in irradiated/implanted materials.

1) The impact on the defect-related physics, and in particular, quantum information science is negligible. In fact, the Authors presented only one example of such behavior and it can be dangerous to draw any general conclusions as we do not know if this behavior is ubiquitous in other systems.

We have never claimed to have contributed to quantum information science in general with this work. We deal with a particular defect which is a good candidate to be used as hardware element in quantum information processing. The scenario we describe here has high relevance for the defect physics of irradiated semiconductors in general. While the latter and quantum optics connect presently, the novelty value of our paper is related to the first, and there, our findings are definitely new. (The four authors have spent altogether almost 100 years on defect physics, and also many weeks on searching for a similar result in the literature.)

The Referee is certainly right, we only investigated one particular case. However, if it happens once, it *can* happen in other cases too. Since scenarios involving the diffusion, transformation, and dissociation of several complexes at the same time often happen in the defect physics of

irradiated/implanted materials, we think it is important to call attention to the possibility. That is why we submitted our paper to a journal with high visibility (and not committed to quantum information science alone).

We note that implantation techniques could create such defect complexes that are otherwise may not form in nature. As a consequence, the number of configurations of defect complexes in a single host material that can be prepared in experiments is astronomical, thus there should be many complexes with similar phenomenon to that we demonstrated on carbon pair formation in silicon, as anticipated from mere statistical point of view. The computational design of defect complexes can accelerate to find the appropriate ones, where *the best* defect complex is often searched for, with the most wanted target properties. That is why we think that our main result is of high interest in the era of introducing machine learning techniques for exploration of defects in solids: The vast majority of defects are complexes, and nowadays machine learning algorithms are conditioned to find the energetically most stable defects for the given applications. Our results on a *single* complex demonstrate that a key complex defect promising in quantum technology, which is one of the most important driving forces in defect physics but definitely not the only one, can be overlooked with this condition.

Actually, we also think, that if the G-center is going to be used in future quantum communication implementations, the mechanism of its creation must be fully understood, in order to work out the know-how for its upscaled synthesis. So we would like to oppose even the statement of the Referee that the importance of our result is negligible in that field.

3) There is a lack of novelty when it comes to methodology. The Authors used very well-established and standard methods to perform the research. It is nothing wrong with using standard computational tools if the research itself carries a significant impact. However, for Nature Communication publication the methodology must be novel, in particular, when the impact of the research itself is negligible or moderate.

Indeed, the methods we use are slowly becoming standard (in a good part because of our earlier work, see e.g. ref. [20] in the original MS), although, as the other Referees pointed out, we use them at quite a high level. We do not think that all papers in Nature Communications must bring methodology development, and, as explained above, the impact of the research is definitively not negligible for defect physics.

4) I am surprised that the Authors did not even discuss the configurational and vibrational entropy contributions to the Gibbs free energy. In fact, these contributions can be significant (the order of 0.2 – 0.5 eV) between various defects (for example Cu in silicon) and at high enough temperature may change the thermodynamic picture. It is also possible that the entropy contributions are very similar for the different structures of C-C defect and cancel each other out when the relative stability is considered. This issue need to be addressed.

Since we are investigating transformations of low symmetry defects into each other, the change of the configurational entropy is certainly negligible. Following up on this comment though, we have made a PBE estimate of the contribution of harmonic vibrations to the Helmholtz free energy at 100°C (where the signal of interstitial carbon disappears and the G-center forms), at the geometry of the A-configuration and at that of its dissociation barrier. We note that these calculations required to recalculate the minimum energy path and get the full vibration spectrum at each point in the path, thus those were not trivial calculations even at PBE level. The results show that the 0K PBE dissociation barrier (without zero-point-energy correction), 1.56 eV, is lowered by only 0.13 eV. Therefore, we can say that temperature effects do not influence our conclusions. In the revised MS and new Supplemental Notes, we explicitly mention this as follows.

A PBE estimate of the contribution of harmonic vibrations to the Helmholtz free energy barrier was obtained at 100°C (see Methods and Supplemental Note 2). It has lowered the barrier energy only by 0.13 eV which is within 7% w.r.t the calculated HSE06 barrier energy at zero kelvin without zero-point energy correction.

REVIEWERS' COMMENTS

Reviewer #1 (Remarks to the Author):

This is my second review of this manuscript. I read carefully the answers from the authors, and the changes made in the manuscript. I now recommend to accept it for publication.

Reviewer #2 (Remarks to the Author):

I would confirm my previous general appraisal on this research: high scientific quality and broad interest. In addition, I have analysed the new version of the manuscript, the detailed response to the reviewers' comments and the current state-of-the art in the field and I would support the publication of the paper in the current form. Indeed, in the revised version the further improvements requested by rev. 1 and rev. 2 have been properly addressed and integrated in the paper. Moreover, I do not agree with the general comment of ref. 3 who criticises the novelty content of this research and the lack advancement of methodology. Indeed, the topic of the paper is new and it brings a due attention on often neglected aspects for ab-initio kind of theoretical analysis. Moreover, the methodological advancements are not among the principal scopes of the journal: the methods should be at the state-of-the-art, but not necessarily novel, in the field since the journal is not a specialised one in the computational material science.

Reviewer #3 (Remarks to the Author):

I read the response provided by Authors. All my objections were properly addressed and now I am in favor of accepting the manuscript in Nature Communications. Congratulations to all Authors.